# Biopsychosocial Factors Associated with Activities of Daily Living Limitations in Chronic Kidney Disease Patients: Insights from the Brazilian Population

**DOI:** 10.3390/ijerph21121680

**Published:** 2024-12-17

**Authors:** Hellen de Carvalho Lima, Joubert Vitor de Souto Barbosa, Adson Aragão de Araújo Santos, Rafael Limeira Cavalcanti, Adirlene Pontes de Oliveira Tenório, Matheus Rodrigues Lopes, Thais Sousa Rodrigues Guedes, Achilles de Souza Andrade, Geronimo José Bouzas Sanchis, Rodrigo Pegado, Johnnatas Mikael Lopes, Marcello Barbosa Otoni Gonçalves Guedes

**Affiliations:** 1Department of Medicine, Federal University of Vale do São Francisco, Paulo Afonso 56304-917, BA, Brazil; hellen@univasf.edu.br (H.d.C.L.); adson@univasf.edu.br (A.A.d.A.S.); adirlene@univasf.edu.br (A.P.d.O.T.); matheus@univasf.edu.br (M.R.L.); johnnatas.lopes@univasf.edu.br (J.M.L.); 2Department of Physiotherapy, Federal University of Rio Grande do Norte, Natal 59078-900, RN, Brazil; jvbsouto@gmail.com (J.V.d.S.B.); rafaellimeirac@hotmail.com (R.L.C.); thais.sousarodrigues@gmail.com (T.S.R.G.); gero.bouzas@gmail.com (G.J.B.S.); rodrigopegado@gmail.com (R.P.); 3Department of Internal Medicine, Federal University of Paraíba, João Pessoa 58051-900, PB, Brazil; achillesandrade@gmail.com

**Keywords:** chronic kidney disease, quality of life, public policy, chronic limitation of activity

## Abstract

Chronic kidney disease (CKD) can impair activities of daily living (ADL), reducing quality of life. The influence of biopsychosocial factors on ADL limitations among CKD patients remains unclear. This study aims to investigate associations between these factors and ADL limitations among CKD patients in the Brazilian population. We analyzed data from 839 individuals diagnosed with CKD obtained from the National Health Survey. The outcome was the presence or absence of limitations in ADL caused by CKD. Biopsychosocial factors included clinical and health status, self-perceived behavior, contextual social support, lifestyle, and household characteristics. Cox regression was employed to adjust interactions between these factors, with the prevalence ratio used as a measure of effect (α ≤ 5%). From the analyzed sample, 373 CKD patients (40.7%; 95% CI: 35.4–46.1%) reported experiencing limitations in ADL. These limitations were associated with individual and contextual factors, including lack of private health insurance, residing in a rural area, poorer self-perceived health, presence of depressive symptoms, physical/mental disabilities, use of medications, and undergoing hemodialysis. Limitations in ADL among individuals with CKD are associated with biopsychosocial factors in the Brazilian population, emphasizing the necessity for public policies that support enhanced therapeutic management and address behavioral health.

## 1. Introduction

Chronic kidney disease (CKD) is traditionally defined as a functional or structural abnormality of the kidneys that persists for at least three months and negatively impacts health [1]. The risks and severity of CKD depend on criteria such as its underlying cause (mainly diabetes and hypertension), the glomerular filtration rate, and the degree of albuminuria [2]. The etiology of CKD remains unclear in many cases, potentially complicating disease screening; however, accurate risk prediction enables early detection and treatment [3,4].

CKD is a global public health problem that significantly elevates the risk of mortality [5]. According to data from the Global Burden of Disease Study (2017), CKD affected nearly 700 million people worldwide (an estimated global prevalence of 9.1%) and resulted in at least 1.2 million deaths that year (the 12th leading cause of death) [6].

In Brazil, there is limited data on the prevalence of CKD [7]. It is estimated that over 10 million people suffer from renal problems in the country [8]. A previous study found a self-reported CKD prevalence of 1.4% in the Brazilian population in 2013 and 2019, with higher rates observed among the elderly [9]. Additionally, the Brazilian Dialysis Survey (2022) reported a prevalence of 758 individuals undergoing dialysis per million population [10].

Given its complexity, CKD prevalence is likely to vary based on biopsychosocial (individual and contextual) factors, including age, sex, ethnicity, lifestyle, comorbidities, socioeconomic status, etc. [11,12]. In Brazil, there is evidence that alterations in key CKD markers, such as glomerular filtration rate and albuminuria levels, increase with age and are higher among Black/Indigenous individuals and those with lower incomes [13].

Allied with these factors, it is important to note that CKD patients exhibit diverse signs and symptoms that affect the quality of life and cognitive/physical functioning, potentially leading to disabilities in activities of daily living (ADL) [14,15] and impacting leisure, work, social activities, and daily routines. Some studies have investigated the relationship between ADL limitations and individual and contextual factors in CKD patients [15,16].

However, literature on this topic is scarce in Brazil, possibly due to the significant ethnic, socioeconomic, and cultural variability across the country, as well as underestimation of CKD diagnosis. Therefore, this study aims to assess the biopsychosocial factors associated with limitations in ADL among individuals with CKD in the Brazilian population.

## 2. Materials and Methods

### 2.1. Study Design

This is a cross-sectional study based on secondary data from the National Health Survey (NHS) conducted in Brazil (2013), implemented by the Brazilian Institute of Geography and Statistics (IBGE) in collaboration with the Oswaldo Cruz Foundation (FIOCRUZ). Participants signed a Free and Informed Consent Form (TCLE) prior to the data collection. Approval was obtained from the National Research Ethics Commission under protocol number 10853812.7.0000.0008. This study adhered to the principles of the Declaration of Helsinki.

### 2.2. Setting

The NHS population consisted of permanent private residents grouped into census tracts in capitals, metropolitan areas, and inland cities. Brazil is a diverse country with a population exceeding 200 million people, characterized by significant socioeconomic, racial, and geographic disparities. The population is predominantly urban, but large rural and Indigenous communities also exist. The survey aimed to capture a representative sample of this diverse population. Excluded from the survey were Indigenous villages, military bases and barracks, camping areas, motels, boats, penitentiaries, penal colonies, prisons, almshouses, orphanages, convents, and hospitals [14]. In addition, individuals experiencing homelessness, prisoners, and Indigenous populations living in villages were also excluded from the sample.

The sample size was calculated based on data from the entire population extracted from the 2010 census conducted by the Brazilian Institute of Geography and Statistics, estimating a sample size of 81,357 individuals aged 18 years and older. From this primary sample, 60,202 respondents were identified, among whom 839 had a diagnosis of CKD.

Sampling was conducted using a conglomerate approach, starting with the selection of census sectors, followed by households within these sectors, and concluding with the selection of individuals within those households. Simple random sampling was employed in the latter two stages [17].

### 2.3. Variables and Data Instrument

The study outcome was the presence or absence of limitations in ADL caused by CKD, assessed by the following question: “In general, to what extent does renal insufficiency limit your daily activities (such as working and performing household tasks, etc.)?”. Participants answered with the options “does not limit”, “a little”, “moderately”, “intensely”, or “very intensely”, which were subsequently dichotomized as “not limited” and “limited”. The choice of this classification was based on methodological considerations for constructing an explanatory model, which takes into account the variability in outcome responses and the number of independent variables.

The independent variables encompassed the following sociodemographic attributes: sex (male/female), age (years), living with a partner (yes/no), color/race (white/nonwhite), social class based on Brazilian criteria, occupation (paid/unpaid), educational level (graduate/postgraduate, high school, primary school, preschool, and illiterate), area of residence (urban/rural), and health insurance status (yes/no).

Data collection took place online, by completing the PNS questionnaire, available through the link: https://www.pns.icict.fiocruz.br/questionarios (accessed on 1 November 2021). The participants first completed a research instrument organized into thematic dimensions from A to W. These dimensions included the analysis of biopsychosocial attributes of the residents, such as demographic and social characteristics, lifestyle, health status, self-perception, chronic diseases, and household characteristics [17].

The main variables under these dimensions included: (a) “health status”: drug use for CKD (yes/no), undergoing hemodialysis (yes/no), undergoing peritoneal dialysis (yes/no), history of kidney transplantation (yes/no), age at CKD diagnosis (years), presence of other noncommunicable diseases (NCDs) (yes/no), presence of physical or mental disability (yes/no), presence of hypertension (yes/no), time since last doctor visit (within the last 12 months, 1–2 years ago, 2–3 years ago, 3 or more years ago), and frequency of family health strategy visits (monthly, less than monthly, or never); (b) “self-perception”: health (good/average/poor), behavioral oscillation (yes/no), and depressive symptoms (yes/no); And “lifestyle”: smoking habits (yes, daily; yes, less than daily; and no current smoker), alcohol use (yes/no), engagement in physical exercises for at least 150 min per week (yes/no), family social support (≤1 person, 2–3 people, and 4 or more people), and friend social support (≤1 person, 2–3 people, and 4 or more people).

It is important to note that the diagnosis of chronic diseases, including CKD, was self-reported by the participants. Research shows that self-reported information on chronic diseases tends to be highly accurate, especially for conditions like CKD, making it a reliable source in epidemiological studies [18].

### 2.4. Statistical Methods

The data analysis relied on a weighted estimation of units from the final stage of sampling. Results were reported as prevalence estimates with a 95% confidence interval (95% CI). Associations between outcomes and independent variables were assessed using the Cox regression model, adjusting for the complex design, and through statistical significance testing with the Wald chi-square test. Additionally, the prevalence ratio (PR) was calculated as a measure of effect, with α set at ≤0.05. The model was constructed hierarchically to control for multicollinearity among the independent variables, ensuring both theoretical and methodological rigor. From an analytical perspective, the potential correlations between independent variables were evaluated using standardized correction indicators. Multicollinearity was ruled out when these indicators were below the threshold of 0.3. Furthermore, potential confounding factors were accounted for by maximizing their control within the adjusted statistical model (independent main effects), and the data were ordered to reflect simultaneous or historically linked events.

## 3. Results

The NHS identified 839 participants with a reported CKD diagnosis, representing approximately 1.5% of the total study population, ranging from 1.3% to 1.7%. Among these participants, 373 (40.7%; 95%CI: 35.4–46.1%) reported some form of limitations in ADL. There were 500 female participants (57.2%; 95%CI: 52.6–61.8%) and 381 (53.1%; 95%CI: 49.1–57.0%) identified as nonwhite. Most of these individuals resided in urban areas (86.9%; 95%CI: 85.8–87.9%), and 57.9% (95%CI: 53.6–62.0%) were not engaged in paid work.

The analysis revealed that being nonwhite (PR = 0.71; 95%CI: 0.55–0.93) and having received formal education (PR = 0.49; 95%CI: 0.29–0.85) were factors less associated with limitations in ADL among individuals with CKD. Additionally, residing in urban areas (PR = 0.60; 95%CI: 0.47–0.75) and being engaged in paid work (PR = 0.67; 95%CI: 0.49–0.89) were also less associated with limitations in ADL compared to their counterparts (Table 1).

Most individuals in the sample used medications for CKD-related comorbidities (57.4%; 95%CI: 51.6–62.0%) and did not undergo hemodialysis (93.0%; 95%CI: 89.7–95.3%) or peritoneal dialysis (98.6%; 95%CI: 97.2–99.3%). It was inferred that individuals undergoing drug treatment and hemodialysis were more associated with limitations in ADL. Peritoneal dialysis and transplantation did not appear to significantly affect individual and social functionality (Table 2).

We found that 82.8% (95%CI: 79.5–85.8%) of individuals had some other chronic health condition, with a 56% higher association with limitations in ADL (PR = 1.56; 95%CI: 1.03–2.38). Additionally, individuals with diabetes (PR = 1.42; 95%CI: 1.01–1.99) and those with physical disabilities (PR = 0.63; 95%CI: 0.45–0.88) were also more associated with limitations in ADL. Conversely, having good (PR = 0.41; 95%CI: 0.28–0.62) and average (PR = 0.59; 95%CI: 0.42–0.85) general health perceptions, as well as not reporting depressive symptoms (PR = 0.62; 95%CI: 0.47–0.81) or emotional oscillation (PR = 0.67; 95%CI: 0.52–0.85), were less associated with limitations in ADL (Table 3).

It was also found that 80% (95% CI: 76.1–83.5%) of individuals did not smoke. However, those with fewer limitations in ADL tended not to adhere to tobacco abstinence and smoked daily (PR = 0.54; 95% CI: 0.36–0.81) or almost daily (PR = 0.07; 95% CI: 0.02–0.24), despite having renal functional changes. Conversely, individuals with limitations in ADL adhered to healthy lifestyle habits, such as not drinking more than once a month, not smoking, and engaging in regular physical activity (PR = 1.52; 95%CI: 1.13–2.05). Most individuals did not have health insurance (65.4%; 95%CI: 61.2–69.3%), and those who did have some form of health insurance showed greater functional status (PR = 0.46; 95%CI: 0.33–0.63) (Table 4).

To understand the relationship of the independent variables with the outcome and control their interactions, it was important to perform an adjusted analysis of the factors related to the interference in the daily activities of CKD patients. This analysis revealed lower levels of limitations in ADL among individuals who reported having good (PR = 0.64; 95%CI: 0.42–0.97) or moderate (PR = 0.72; 95%CI: 0.55–0.94) self-perceived health, absence of depressive symptoms (PR = 0.72; 95%CI: 0.55–0.94) or physical disabilities (PR = 0.62; 95%CI: 0.47–0.82), as well as those living in urban areas (PR = 0.75; 95%CI: 0.60–0.94) and those who have private health insurance (PR = 0.51; 95%CI: 0.35–0.73). Furthermore, individuals who smoked daily (PR = 0.36; 95%CI: 0.20–0.64) or almost daily (PR = 0.05; 95%CI: 0.01–0.17) showed associations with lower levels of limitations in ADL. Conversely, those who did not receive drug treatment for CKD-related comorbidities (PR = 1.40; 95%CI: 1.09–1.80) and individuals undergoing hemodialysis (PR = 2.30; 95%CI: 1.74–3.03) were associated with higher levels of limitations in ADL (Table 5).

## 4. Discussion

Our main results indicated that CKD patients experienced less impact on ADL when they had a good or average perception of health, lived in urban areas, and had private health insurance. In contrast, physical or intellectual disabilities, ongoing drug use, and undergoing hemodialysis were linked to greater ADL limitations. We also observed that race, age, sex, and employment status were not associated with limitations in ADL despite CKD being more common in individuals aged 30 to 50 years.

A lower educational level was linked to greater ADL limitations due to its impact on behavior and health. People with lower educational levels often face economic vulnerabilities, leading to higher rates of sedentary lifestyle, obesity, alcohol use, and smoking, which increase the prevalence of NCDs like CKD [17]. These conditions worsen social inequalities, especially among nonwhite individuals, affecting access to healthcare and complicating participation in health education and prevention programs [19].

Another sociodemographic factor directly associated with ADL limitations among Brazilian individuals with CKD is their area of residence. Urban residents experienced fewer limitations in ADL compared to rural residents. This difference may occur because rural populations often prioritize agricultural work, which may obscure chronic health conditions and reduce healthcare-seeking behavior [20]. Individuals with CKD living in rural areas tend to experience greater debility due to the progressive nature of the disease, and their distance from urban centers exacerbates the difficulty in accessing essential services for physical rehabilitation, nutritional care, and psychological support. These services often depend on reliable public or private transportation, which can be limited or unavailable in rural areas, further complicating care. Additionally, most of Brazil’s rural population is of low income, which may present a potential analytical interaction with the factors investigated in this study, as financial constraints can limit access to transportation and healthcare services. Limited access to healthcare facilities is another significant factor, as urban areas tend to concentrate on these services, necessitating travel for rural residents. Addressing these disparities requires local and regional public policies targeting rural populations to ensure equitable healthcare access and promote early diagnosis [21].

Interestingly, we observed that patients who engage in continuous drug use tend to experience greater limitations in ADL compared to non-users. It is important to interpret this finding carefully, as it does not necessarily imply that drugs worsen functionality. Rather, CKD is often asymptomatic in its early stages, leading to poor adherence to therapy until the disease progresses. Inadequate information and communication with patients can contribute to therapy non-adherence, especially early in the disease course [22]. Individuals using these drugs may be at more advanced stages of CKD, thus experiencing more limitations. It is noteworthy that non-adherence to therapy due to drug use can worsen disease progression [23].

It was also observed that hemodialysis is associated with greater limitations in ADL. Hemodialysis brings about various physical consequences, including arterial hypotension, nausea, dizziness, cramps, headache, and fainting [24]. These effects significantly impact the quality of life and have personal, familial, and social repercussions. CKD disrupts daily activities such as diet, travel, eating habits, work, and social life, resulting in dependence on dialysis sessions for affected individuals [25]. Despite these complications, dialysis remains one of the safest and most effective procedures for replacing renal function and improving the quality of life for individuals with CKD. Peritoneal dialysis offers advantages by reducing individual and social restrictions, as it can be performed at home with minimal disruption to daily routines. This option is increasingly promoted through specific policies aimed at enhancing patient quality of life [26]. Greater social support for contextual adjustments and emphasis on patient education in renal replacement therapy likely play key roles in the decision-making process between hemodialysis and peritoneal dialysis [27]. Moreover, peritoneal dialysis is more cost-effective for the healthcare system [28,29,30].

Another key factor is the perception of depressive symptoms, which are observed in just over half of the CKD population, aligning with findings from other studies. CKD necessitates specific treatment and requires adaptations and lifestyle changes, potentially increasing the likelihood of anxiety and depressive symptoms among affected individuals. Moreover, these necessary adjustments can lead to social isolation and a decline in functional, physical, or mental performance, directly impacting their lives [31].

Self-perception of health serves as a valuable indicator for assessing individual health, encompassing its physical, mental, and social dimensions [32]. Moreover, it reflects the effectiveness of health education initiatives aimed at improving societal health outcomes and lifestyles. Adjusted analysis of the results has revealed an inverse relationship: individuals with a better perception of their health tend to experience fewer limitations in ADL. Tracking health perception could, therefore, serve as a useful tool in health services for identifying individuals at risk of clinical and functional decline, as well as mortality [33].

Furthermore, individuals with private health insurance were able to mitigate limitations in ADL among CKD patients. Private health insurance is often associated with higher social status, improved living conditions, and better access to healthcare services. The private healthcare system enhances flexibility and expedites examinations and procedures, offering convenience and easier access for those who can afford it [34]. Consequently, this results in reduced complications and limitations caused by the disease. In Brazil, most hemodialysis services are provided by private entities, which allocate beds funded through both public and private resources. This creates a significant disparity, as individuals with private health insurance face fewer barriers to accessing hemodialysis services, given that the competition for available slots is much lower and governed by more structured commercial regulations [35]. Consequently, this results in reduced complications and limitations caused by the disease.

An intriguing finding emerged from the adjusted analysis of smoking habits: we found that individuals who smoke tend to have fewer limitations in ADL compared to non-smokers. This pattern parallels that observed with drug use, suggesting that individuals with CKD who smoke may be in the early stages of the disease or have not yet experienced significant health consequences from smoking [36]. Conversely, non-smokers may have quit smoking due to more severe stages of CKD and impaired health status. Based on these observations, early educational and support initiatives are crucial to encourage tobacco reduction and delay the onset of limitations associated with this habit.

Additionally, individuals with CKD experience increased limitations in ADL when they also have physical disabilities. These limitations affect their daily habits and routines, interfere with tasks such as accessing healthcare facilities and undergoing therapeutic interventions, and impact their emotional well-being [37].

Finally, it is important to acknowledge several limitations of this study. Notably, the cross-sectional design prevents the establishment of causal relationships between variables. Additionally, outcomes were assessed using self-reports instead of official medical diagnoses. While self-reporting is highly sensitive and effectively captures many conditions, it may still have some limitations. Another important limitation of this study is the lack of detailed information about the stage of chronic kidney disease (CKD), which prevents specific control of this factor. Although other chronic diseases coexisting with CKD were jointly controlled under the “noncommunicable diseases” (NCD) variable in Table 3 and Table 5, these conditions were not relevant in the adjusted model.

## 5. Conclusions

Individuals with CKD experience limitations in ADL primarily due to psychosocial factors such as smoking, educational level, access to health services, and the type of care received. Public health policies must address these factors when organizing health services and systems to mitigate the reduction in quality of life, improve treatment adherence, and prevent the worsening of clinical conditions. Failure to do so could increase costs to the healthcare system.

## Figures and Tables

**Table 1 ijerph-21-01680-t001:** Descriptive and unadjusted analysis of sociodemographic and contextual factors.

	N (%)	95% CI	β	t	*p*	PR_u_	95% CI
Sex							
Male	339 (42.8%)	38.2–47.4	−0.15	−1.04	0.29	0.85	0.63–1.14
Female	500 (57.2%)	52.6–61.8	0			1	
Age (years)	53.74 *	52.21–55.27	0.001	0.12	0.89	1.001	0.99–1.10
Color or race							
White	381 (53.1%)	49.1–57.0	−0.33	−2.47	0.01	0.71	0.55–0.93
Nonwhite	458 (46.9%)	43.0–50.9	0			1	
Lives with a partner							
Yes	470 (70.3%)	66.4–73.8	−0.25	−2.28	0.02	0.77	0.62–0.96
No	369 (29.7%)	26.2–33.6	0			1	
Education level							
Graduation/Post-graduation	114 (11.4%)	9.2–14.0	−0.37	−1.21	0.22	0.68	0.37–1.25
High school	198 (24.1%)	20.6–27.9	−0.91	−2.87	0.004	0.39	0.21–0.74
Primary school	244 (29.4%)	25.9–33.1	−0.76	−2.51	0.01	0.46	0.25–0.84
Preschool	204 (25.9%)	22.6–29.6	−0.61	−1.98	0.04	0.53	0.29–0.99
Illiterate	79 (9.25%)	6.8–12.4	0			1	
Health insurance							
Yes	217 (34.6%)	30.7–38.8	−0.78	−4.88	<0.001	0.46	0.33–0.63
No	622 (65.4%)	61.2–69.3	0			1	
Area of residence							
Urban	676 (86.9%)	85.8–87.9	−0.51	4.39	<0.001	0.60	0.47–0.75
Rural	163 (13.1%)	12.1–14.2	0			1	
Paid work							
Yes	355 (42.1%)	38.0–46.4	−0.40	2.68	0.008	0.67	0.49–0.89
No	484 (57.9%)	53.6–62.0				1	
Social class	19.45 *	18.76–20.15	−0.34	−3.28	0.001	0.70	0.57–0.87

Legend: * Continuous variable presented as a mean; N—number of participants; %—ratio; CI—Confidence interval; β—Unadjusted regression coefficient; t—*t* test for the comparison between levels; *p*—Statistical significance; PR_u_—Unadjusted prevalence ratio.

**Table 2 ijerph-21-01680-t002:** Description of drug use, hemodialysis, peritoneal dialysis, and kidney transplant in the Brazilian population with CKD.

	N (%)	95% CI	β	t	*p*	PR_u_	95% CI
Age at CKD diagnosis	40.26 *	38.26–42.26	0.01	0.82	0.41	1.00	0.99–1.01
Drug use							
Yes	475 (57.4)	51.6–62.09	0.58	4.67	<0.001	1.79	1.40–2.29
No	364 (42.6)	37.1–48.4	0			1	
Hemodialysis							
Yes	56 (7.0)	4.7–10.3	0.88	4.72	0.001	2.41	1.67–3.48
No	783 (93.0)	89.7–95.3	0			1	
Peritoneal dialysis							
Yes	12 (1.4)	0.7–2.8	0.39	1.11	0.27	1.48	0.74–2.98
No	827 (98.6)	97.2–99.3	0			1	
Kidney transplant							
Yes	17 (1.9)	1.0–3.3	0.13	0.38	0.70	1.14	0.57–2.29
No	822 (98.1)	96.7–99.0	0			1	

Legend: * Continuous variable presented as a mean; N—number of participants; %—ratio; CI—Confidence interval; β—Unadjusted regression coefficient; t—*t* test for the comparison between levels; *p*—Statistical significance; PR_u_—Unadjusted prevalence ratio.

**Table 3 ijerph-21-01680-t003:** Descriptive and unadjusted analysis of health and self-perception factors.

	N (%)	95% CI	β	t	*p*	PR_u_	95% CI
Other associated NCDs							
Yes	680 (82.8%)	79.5–85.8	0.45	2.09	0.04	1.56	1.03–2.38
No	125 (17.2%)	14.2–20.5	0			1	
Diabetes							
No	650 (83.9%)	80.7–86.7	−0.35	2.07	0.03	0.70	0.50–0.98
Yes	124 (16.1%)	13.3–19.3	0			1	
Hypertension							
Normal	637 (75.0%)	70.4–79.0	−0.26	−1.67	0.10	0.77	0.56–1.05
Hypertense	199 (25.0%)	21.0–29.6	0			1	
Disabilities							
No	657 (73.4%)	68.4–77.9	−0.46	−2.68	0.01	0.63	0.45–0.88
Yes	182 (26.6%)	22.1–31.6	0			1	
Health self-perception							
Good	273 (34.3%)	30.5–38.4	−0.88	−4.27	<0.01	0.41	0.28–0.62
Average	403 (47.1%)	42.7–51.4	−0.52	−2.89	<0.01	0.59	0.42–0.85
Poor	163 (18.6%)	15.5–22.1	0			1	
Perception of mood oscillation							
No	472 (59.1%)	54.3–63.7	−0.40	−3.22	<0.01	0.67	0.52–0.85
Yes	367 (40.9%)	36.3–45.7	0			1	
Perception of depressive symptoms							
No	376 (48.1%)	43.4–52.9	−0.48	−3.47	<0.01	0.62	0.47–0.81
Yes	463 (51.9%)	47.1–56.6	0			1	

Legend: N—number of participants; %—ratio; CI—Confidence interval; β—Unadjusted regression coefficient; t—*t* test for the comparison between levels; *p*—Statistical significance; PR_u_—Unadjusted prevalence ratio; NCDs: noncommunicable diseases.

**Table 4 ijerph-21-01680-t004:** Descriptive and unadjusted analysis of lifestyle factors.

	N (%)	95% CI	β	t	*p*	PR_u_	95% CI
Regular physical activity							
Sedentary	733 (85.6%)	82.2–88.4	0.44	1.61	0.10	1.55	0.90–2.66
Active	106 (14.4%)	11.6–17.8	0		1		
Smoking habit							
Yes, daily	121 (18.4%)	15.1–22.3	−0.62	−3.01	<0.01	0.54	0.36–0.81
Yes, less than daily	14 (1.6%)	0.8–3.0	−2.71	−4.20	<0.01	0.07	0.02–0.24
Do not smoke	704 (80.0%)	76.1–83.5	0			1	
Family social support							
≤1 people	370 (39.1%)	34.8–43.6	0.09	0.55	0.58	1.09	0.79–1.51
2–3 people	246 (32.4%)	28.0–37.1	−0.01	−0.04	0.97	0.99	0.68–1.45
4 or more people	223 (28.5%)	24.7–32.8	0			1	
Friend social support							
≤1 people	520 (59.8%)	54.7–64.7	−0.15	−0.94	0.35	0.86	0.63–1.17
2–3 people	178 (22.0%)	17.9–26.8	−0.25	−1.03	0.30	0.78	0.48–1.25
4 or more people	141 (18.2%)	15.1–21.8	0			1	
Last doctor visit							
Last 12 months	717 (87.3%)	84.6–89.6	0.42	1.40	0.16	1.52	0.85–2.73
From 1 to 2 years	52 (5.8%)	4.0–8.2	−0.03	−0.07	0.94	0.97	0.44–2.13
From 2 to 3 years	31 (2.2%)	1.6–3.1	<0.01	0.01	0.99	1.00	0.47–2.15
3 or more years	39 (4.8%)	3.5–6.5	0			1	
Family health team visit in the last year?							
Monthly	210 (41.7%)	36.0–47.5	−0.16	−1.01	0.32	0.85	0.61–1.17
Less than monthly	199 (42.1%)	36.2–48.3	0.07	0.26	0.79	1.08	0.62–1.87
Never	98 (16.2%)	13.5–19.3	0			1	

Legend: N—number of participants; %—ratio; CI—confidence interval; β—Unadjusted regression coefficient; t—*t* test for the comparison between levels; *p*—Statistical significance; PR_u_—Unadjusted prevalence ratio.

**Table 5 ijerph-21-01680-t005:** Adjusted analysis of factors related to the interference in ADL in CKD patients in the Brazilian population.

Limitations in ADL	PR_adjus_	95% CI
Color		
White	0.92	0.70–1.21
Nonwhite	1	
Lives with a partner		
Yes	0.91	0.74–1.12
No	1	
Education level		
Graduation/Post-Graduation	1.37	0.85–2.20
High school	0.68	0.42–1.11
Primary school	0.60	0.38–0.94
Preschool	0.66	0.41–1.07
Illiterate	1	
Social class	0.82	0.62–1.07
NCDs		
Yes	1.04	0.66–1.62
No	1	
Diabetes		
Yes	1.11	0.82–1.50
No	1	
Physical disability		
No	0.62	0.47–0.82
Yes	1	
Health perception		
Good	0.64	0.42–0.97
Average	0.72	0.55–0.94
Poor	1	
Perception of depressive symptoms		
No	0.72	0.55–0.94
Yes	1	
Perception of mood lability		
No	0.86	0.66–1.12
Yes	1	
Smoking habit		
Daily	0.36	0.20–0.64
Non-daily	0.05	0.01–0.17
Does not smoke	1	
Drugs for CKD-related comorbidities		
Yes	1.40	1.09–1.80
No	1	
Hemodialysis		
Yes	2.30	1.74–3.03
No	1	
Health insurance		
Yes	0.51	0.35–0.73
No	1	
Paid work		
Yes	1.01	0.74–1.37
No	1	
Area of residence		
Urban	0.75	0.60–0.94
Rural	1	

Legend: PR_adju_**_s_**—Adjusted Prevalence ratio; CI—Confidence interval.

## Data Availability

This study used secondary data from the National Health Survey, implemented by the Brazilian Institute of Geography and Statistics in collaboration with the Oswaldo Cruz Foundation, which can be accessed at the following link: https://www.ibge.gov.br/estatisticas/sociais/saude/29540-2013-pesquisa-nacional-de-saude.html?edicao=9161&t=resultados (accessed on 1 February 2024).

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
