# Peer review of "Biopsychosocial Factors Associated with Activities of Daily Living Limitations in Chronic Kidney Disease Patients: Insights from the Brazilian Population"

_ijerph, 2024, doi:10.3390/ijerph21121680_

Round 1
Reviewer 1 Report
Comments and Suggestions for Authors
This article conducts an empirical analysis of the relationship between biopsychosocial factors and the daily living activities of patients with chronic kidney disease (CKD) in the Brazilian population. It explores the association between limitations in daily living activities (ADL) among CKD patients and various biopsychosocial factors, which holds significant clinical and public health value.
Firstly, the choice of topic is aligned with practical needs, addressing the increasing prevalence of CKD and its substantial impact on quality of life. Investigating the factors related to ADL limitations in these patients is of great relevance and innovation. The study analyses data from a national health survey involving 839 CKD patients, taking into account multiple biopsychosocial factors, including health insurance, place of residence, and mental health status, thereby providing a comprehensive perspective on the life challenges faced by CKD patients. This multidimensional analysis not only enriches the current research perspective but also lays the groundwork for formulating more targeted public health policies.
Methodologically, the authors employ Cox regression models to analyse the data, adequately adjusting for various biopsychosocial factors to ensure the scientific validity and reliability of the results. By quantifying the impact of each factor on ADL limitations using prevalence ratios, the study clarifies the importance of different factors. However, despite the rigorous methodology, the paper does not sufficiently explain how it addresses multicollinearity and potential confounding factors, which may somewhat affect the precision of the results and the comprehensiveness of the interpretations.
The findings indicate that ADL limitations in CKD patients are closely related to various factors, particularly the lack of private health insurance, residing in rural areas, poor self-perceived health status, depressive symptoms, and physical or mental disabilities. These findings underscore the need for enhanced health management and psychological support for vulnerable groups in public health policies. However, the paper provides only a superficial discussion on why patients in rural areas experience more severe ADL limitations and the role of private health insurance in alleviating these limitations. A deeper analysis of the mechanistic details surrounding these issues would aid in developing more effective intervention measures.
Overall, the paper is content-rich, employs reasonable analytical methods, and presents findings with high clinical applicability and public health significance. Although there are shortcomings in discussing certain key issues, the authors demonstrate strong research capabilities and logical reasoning. If future research can delve deeper into these critical issues, the impact and practicality of the study will be further enhanced.
Author Response
Comments1: Methodologically, the authors employ Cox regression models to analyse the data, adequately adjusting for various biopsychosocial factors to ensure the scientific validity and reliability of the results. By quantifying the impact of each factor on ADL limitations using prevalence ratios, the study clarifies the importance of different factors. However, despite the rigorous methodology, the paper does not sufficiently explain how it addresses multicollinearity and potential confounding factors, which may somewhat affect the precision of the results and the comprehensiveness of the interpretations.
Response: We appreciate the proposed suggestions and agree with them. To clarify the text, we added information regarding multicollinearity among the independent variables. From an analytical perspective, the potential correlation between independent variables is assessed using standardized correction indicators, with multicollinearity ruled out when these indicators are below 0.3. Similarly, potential confounding factors are controlled for by maximizing adjustments in the statistical model (independent main effects) and by organizing the data to reflect simultaneous or historically linked events. This information can be found in the data analysis section.
Comments 2: The findings indicate that ADL limitations in CKD patients are closely related to various factors, particularly the lack of private health insurance, residing in rural areas, poor self-perceived health status, depressive symptoms, and physical or mental disabilities. These findings underscore the need for enhanced health management and psychological support for vulnerable groups in public health policies. However, the paper provides only a superficial discussion on why patients in rural areas experience more severe ADL limitations and the role of private health insurance in alleviating these limitations. A deeper analysis of the mechanistic details surrounding these issues would aid in developing more effective intervention measures.
Response: Thank you for the suggestion. To provide more comprehensive information, we added to the eighth paragraph of the discussion that most dialysis services in Brazil are provided by private companies, which allocate beds funded by both public and private resources. As a result, access to dialysis services for those with private insurance presents fewer barriers since the competition for available spots is significantly smaller and subject to more commercial regulation.

Reviewer 2 Report
Comments and Suggestions for Authors
Dear authors:
I reviewed your paper entitled “Biopsychosocial factors associated with activities of daily living (ADL) limitations in chronic kidney disease (CKD) patients: insights from the Brazilian population”. The study aims: to investigate associations between these factors and ADL limitations among CKD patients in the Brazilian population. It is a cross-sectional study, that analysed data from 839 individuals diagnosed with CKD, obtained from National Health Survey. From the analysed sample, 373 CKD patients (40.7%; 95% CI: 35.4-46.1%) reported experiencing limitations in ADL.
First, I would like to congratulate all the authors on their submitted paper. It was easy to read and very interesting. As an outsider, my perspective allows me to offer some suggestions and points for review:
In abstract: I suggest inserting the type of study;
- In study design point (page 2), you mention the National Research Ethics Commission, but did you obtain a consent document from the participants?
- In page 2 (point 2. Materials and methods): please considered to insert in “setting” point the information about “sample and population”, because the following paragraphs are all about population/sample and not setting; Please let us know more information about the sample (i.e.: exclusion criteria);
- Please considered change the title “Outcomes and Independent Variables” to “variables and Data instrument” and compiled with information from “Data sources and measurement”;It is not clear how did you applied the instrument? (Online? Presential? And who did apply it?)
- - I recommend using critical appraisal tools to help you complete the missing information. Tools from EQUATOR or JBI could be particularly useful (e.g., https://jbi.global/critical-appraisal-tools).
I have nothing more to add or suggest, and I wish you good luck towards publishing your paper!
Best regard.
Author Response
Comments 1: In study design point (page 2), you mention the National Research Ethics Commission, but did you obtain a consent document from the participants?
Response 1: Thank you for the suggestion. The Study Design section has been revised, and information about participant consent has been added.
Comments 2: In page 2 (point 2. Materials and methods): please considered to insert in “setting” point the information about “sample and population”, because the following paragraphs are all about population/sample and not setting; Please let us know more information about the sample (i.e.: exclusion criteria)
Response 2: Thank you for the suggestions. General information about the Brazilian population has been added to the Setting section to provide context. Additionally, information about the participants excluded from the study has been included.
Comments 3: Please considered change the title “Outcomes and Independent Variables” to “variables and Data instrument” and compiled with information from “Data sources and measurement”;It is not clear how did you applied the instrument? (Online? Presential? And who did apply it?
Response 3: Thank you for the comments. All suggestions have been incorporated, and the changes can be seen in the Variables and Data Instrument section.

Reviewer 3 Report
Comments and Suggestions for Authors
This study assessed the biopsychosocial factors associated with activities of daily living limitations in chronic kidney disease patients in Brazil. Overall, the study is interesting. Howeer, I have several concerns.
1. Please defined the diagnosis of CKD applied in this study.
2. Some confounding factors, such as the CKD stage, underlying comorbidites including stroke, cardiovascular disease, chronic lung disease and diabetes mellitus should be added.
Author Response
Comments 1: Please defined the diagnosis of CKD applied in this study.
Response 1: Thank you for the suggestion. The measurement section has been adjusted.
Comments 2: Some confounding factors, such as the CKD stage, underlying comorbidites including stroke, cardiovascular disease, chronic lung disease and diabetes mellitus should be added.
Response 2: Thank you for the suggestion. The study limitations, located in the discussion section of this article, have been revised.

Round 2
Reviewer 2 Report
Comments and Suggestions for Authors
Dear Authors,
Thank you for your reply and the revised paper.
I have no further comments and wish you the best of luck with its publication!
Best regards.
Reviewer 3 Report
Comments and Suggestions for Authors
The authors resposne well.